# Hybrid Genomic Analysis of *Salmonella enterica* Serovar Enteritidis SE3 Isolated from Polluted Soil in Brazil

**DOI:** 10.3390/microorganisms11010111

**Published:** 2022-12-31

**Authors:** Danitza Xiomara Romero-Calle, Francisnei Pedrosa-Silva, Luiz Marcelo Ribeiro Tomé, Thiago J. Sousa, Leila Thaise Santana de Oliveira Santos, Vasco Ariston de Carvalho Azevedo, Bertram Brenig, Raquel Guimarães Benevides, Thiago M. Venancio, Craig Billington, Aristóteles Góes-Neto

**Affiliations:** 1Postgraduate Program in Biotechnology, State University of Feira de Santana (UEFS), Av. Transnordestina S/N, Feira de Santana 44036-900, BA, Brazil; 2Molecular and Computational Biology of Fungi Laboratory, Department of Microbiology, Instituto de Ciências Biológicas, Universidade Federal de Minas Gerais, Belo Horizonte 31270-901, MG, Brazil; 3Department of Biological Sciences, Feira de Santana State University (UEFS), Feira de Santana 44036-900, BA, Brazil; 4Laboratory of Chemistry, Function of Proteins and Peptides, Center for Biosciences and Biotechnology, Darcy Ribeiro North Fluminense State University (UENF), Campos dos Goytacazes 28013-602, RJ, Brazil; 5Laboratory of Cellular and Molecular Genetics, Department of Genetics, Ecology and Evolution, Institute of Biological Sciences, Federal University of Minas Gerais, Belo Horizonte 31270-901, MG, Brazil; 6Institute of Veterinary Medicine, Burckhardtweg, University of Göttingen, 37073 Göttingen, Germany; 7Health & Environment Group, Institute of Environmental Sciences and Research, P.O. Box 29-181, Christchurch 8540, New Zealand

**Keywords:** whole genome sequencing, *Salmonella*, hybrid sequence assembly, heavy metal, antimicrobial resistance

## Abstract

In Brazil, *Salmonella enterica* serovar Enteritidis is a significant health threat. *Salmonella enterica* serovar Enteritidis SE3 was isolated from soil at the Subaé River in Santo Amaro, Brazil, a region contaminated with heavy metals and organic waste. Illumina HiSeq and Oxford Nanopore Technologies MinION sequencing were used for de novo hybrid assembly of the *Salmonella* SE3 genome. This approach yielded 10 contigs with 99.98% identity with *S. enterica* serovar Enteritidis OLF-SE2-98984-6. Twelve *Salmonella* pathogenic islands, multiple virulence genes, multiple antimicrobial gene resistance genes, seven phage defense systems, seven prophages and a heavy metal resistance gene were encoded in the genome. Pangenome analysis of the *S. enterica* clade, including *Salmonella* SE3, revealed an open pangenome, with a core genome of 2137 genes. Our study showed the effectiveness of a hybrid sequence assembly approach for environmental *Salmonella* genome analysis using HiSeq and MinION data. This approach enabled the identification of key resistance and virulence genes, and these data are important to inform the control of *Salmonella* and heavy metal pollution in the Santo Amaro region of Brazil.

## 1. Introduction

Salmonellosis, one of the primary causes of foodborne infections resulting from gram-negative enteropathogenic bacteria *Salmonella* spp., is a global threat to human health [1]. Typhoidal *Salmonella* causes enteric fever in humans, whereas non-typhoidal *Salmonella* (NTS) results in acute/chronic gastroenteritis. Annually, it is estimated that NTS is responsible for ~93.8 million infections and ~155,000 deaths [2]. 

NTS infections cause diarrhoea and a non-specific febrile illness that is clinically indistinguishable from other febrile illnesses [3]. *Salmonella enterica* subspecies *enterica* has more than 2600 serovars according to unique somatic (O) and flagellar (H) antigenic formulae [4,5]. *S. enterica* sv. Typhimurium and *S. enterica* sv. Enteritidis are the main pathogens responsible for causing gastroenteritis in humans [6,7].

To prevent the occurrence of the main *Salmonella* serovars worldwide, several prevention and control measures are adopted in farms and food processing industries. In Brazil, *Salmonella* infection of flocks and transmission to poultry-derived food is a major transmission route for the pathogen. *Salmonella* is routinely managed on Brazilian farms by poultry vaccination and laboratory testing (Available online: https://www.gov.br/agricultura/pt-br/assuntos/sanidade-animal-e-vegetal/saude-animal/programas-desaude-animal/pnsa/2003_78.INconsolidada.pdf (accessed on 18 December 2022)). However, despite these measures several poultry diseases and foodborne *Salmonella* outbreaks have been reported in Brazil in recent decades [8].

Whole-genome sequencing (WGS) is useful in foodborne outbreak investigations and pathogen surveillance [9]. Illumina short-read sequencing technology has proven to be robust for characterizing pathogens of clinical care [10], but it is unable to resolve repetitive and GC-rich regions, thus producing unresolvable regions in the underlying genome assembly [11]. These unresolved regions impede completion of a whole-genome structure, which is crucial to determine if some genes are co-regulated or co-transmissible, and if they are located on the chromosome or plasmids [12]. Furthermore, the bias to identify key virulence genes during an outbreak investigation can also have negative impacts on public health assessment. 

Nanopore sequencing technology can generate long reads to facilitate the completion of bacterial genome assemblies but can lack sequencing depth in some repetitive regions [13]. However, nanopore’s long reads can span wide repetitive regions and help solve GC-rich regions, making it useful for resolving full-length genome sequences [14]. Nanopore sequencing technology exhibits lower read accuracy than Illumina sequencing which can produce systematic errors, as a result, it has only usually been applied as a complement to short-read sequencing for bacterial genome assembly [15]. Since the release of the MinION platform by Oxford Nanopore Technologies, nanopore chemistry, base-calling, and bioinformatic tools have been steadily improving and are now more able to produce accurate bacterial genome sequences independent of other sequencing technologies [16].

The combination of both short reads for base-calling accuracy and long reads for structural integrity has recently been developed as a hybrid assembly approach to close whole-genome assemblies, such as those found in the Unicycler and SPAdes pipelines [17,18]. Unicycler was specifically developed for hybrid assembly of bacterial genomes [18]. Unicycler generates a short-read assembly graph and then uses long-reads to build bridges to resolve all repeats in the genome, performs multiple rounds of short-read polishing and finally, it produces a complete genome assembly [14].

In this study, a hybrid genome assembly approach using MinION and HiSeq sequencing data was used to improve the assembly parameters and gene completeness, identification of virulence and antimicrobial resistance genes (ARG), genome phylogeny and pangenome in *Salmonella enterica* var. Enteritidis SE3 isolated from soil at the Subaé river in Santo Amaro, Brazil, a river polluted with organic waste and heavy metals.

## 2. Materials and Methods

Environmental soil samples were obtained from the Subaé river basin in Santo Amaro, Salvador de Bahia, Brazil. Approximately 100 g of soil sample was collected from river soil (12°31′46.77″ S 38°44′1.24″ W). The sample was transported in a refrigerated box (4–8 °C) to the laboratory where the analyses were undertaken immediately.

### 2.1. Salmonella Isolation

*Salmonella* was isolated according to the US Food and Drug Administration Bacteriological Analytical Manual (https://www.fda.gov/media/79991/download (accessed on 18 December 2022)). Briefly, 10 g or 10 mL of samples of each sample were pre-enriched in 100 mL lactose broth (supplier), at 37 °C for 24 h, 0.1 mL of pre-enriched culture was transferred to 10 mL enriched in Tetrathionate (TT) broth (HIMEDIA, Kennett Square, PA, USA) and incubated at 41 °C for 24 h. Broth cultures from the selective enrichment broth were plated on Xylose-Lysine-Desoxycholate (XLD) agar (HIMEDIA, Kennett Square, PA, USA), Bismuth sulfite agar (Acumedia Manufacters Inc., San Bernardino, CA, USA) and Salmonella Shigella (SS) agar (HIMEDIA, PA, Kennett Square, USA). Colonies characteristic of *Salmonella* having a slightly transparent zone of reddish color and a black center for XLD, gray or brown-black colonies with or without metallic sheen for Bismuth Sulfite Agar, and beige colonies with black centers for SS agar were identified and picked. Then, the isolates were tested biochemically using the Triple Sugar Iron (TSI) test. *Salmonella* strains were confirmed when they showed good to excellent growth, pink colonies with black centers were detected, and the agar was red [19]. 

### 2.2. DNA Isolation

For bacteria, a single colony was enriched in 5 mL Luria Bertani (LB) broth, and 15 mL of enrichment broth was transferred to a centrifuge tube and centrifuged at 4000 rpm for 10 min. DNA from *Salmonella* strains was extracted and purified using the E.Z.N.A. Bacterial DNA Mini Kit (Omega Biotek, Norcross, GA, USA) following the instructions provided by the manufacturer. For phages, a crude lysate was centrifuged the lysate as described. DNA isolation from phages was carried out using the E.Z.N.A. Viral DNA Mini Isolation Kit (Omega Biotek, Norcross, GA, USA) following the instructions provided by the manufacturer. The quality and concentration of the bacteria and phage DNA was evaluated by Qubit Fluorometric Quantification (ThermoFisher Scientific, Waltham, MA, USA) and gel electrophoresis (1% of agarose gel, 80 V for 45 min in 1x TAE Buffer).

### 2.3. Amplification of 16S rRNA Gene

PCR amplification was performed using a Veriti^TM^ 96-Well Thermal Cycler (Applied Biosystems, Foster City, CA, USA), 16S gene Amplification PCR for the amplification of the 16S rRNA gene was carried out using universal primers 27F (5′-AGAGTTTGATCATGGCTCAG-3′) as forward and 1492R (5′-GGTTACCTTGTTACGACTT-3′) as a reverse primer [20]. Approximately 10–100 ng of template was added to a reaction mix containing 10 μL Master Mix 2x (Qiagen, Germantown, MD, USA), 1 μL primer 27 F (10 μM), 1 μL primer 1492R (10 μM), and 1 μL reverse primer (10 μM). PCR was performed with the following cycling conditions: initial denaturation at 95 °C for 10 min, 35 cycles of denaturation at 95 °C for 1 min, annealing from 50 °C to 60 °C for 1 min, and extension at 72 °C for 1 min. A final extension was performed at 72 °C for 7 min. PCR products were visualized using GelRed (Biotium, San Francisco, CA, USA) on a 2% agarose gel which had been run at 80 V for 30 min. The separated PCR products were visualized under UV light and photographed.

### 2.4. 16S rRNA Gene Sequencing and Phylogenetic Analysis

The amplified 16S rRNA PCR products were purified and sequenced at Macrogen (Seoul, Republic of Korea) using the ABI 3100 sequencer with Big Dye Terminator Kit v.3.1. The same 16S rRNA primer sequences used for PCR were used for sequencing. The sequences were assembled and trimmed using Geneious Prime and submitted to the Greengenes database (https://rnacentral.org/expert-database/greengenes, accessed on 18 December 2022). The sequences of this study and reference sequences were aligned with Clustal W, and the evolutionary history was inferred using the Neighbor Joining Method [21] and the percentage of replicate trees in which the associated taxa clustered together in the bootstrap test (1000 replicates). There were a total of 1552 positions in the final dataset. Evolutionary analyses were conducted in MEGA X [21].

### 2.5. Whole Genome Sequencing (WGS) by MinION and Illumina

Nanopore WGS sequencing was carried out at the Molecular and Computational Biology of Fungi Laboratory, Federal University of Minas Gerais (UFMG). The DNA library was prepared with ligation Sequencing kit (SQKRAD004, Oxford Nanopore Technologies, Oxford, UK) according to the manufacturer’s instructions. Libraries were sequenced with qualified FLO-MIN106 flow cells (the initial bias voltage was −210 mV and the active pores number around 516) for 36 h (basecalling function was used, the reads sequences was filtered using a min_score = 9) on a MinION (Oxford Nanopore Technologies, Oxford, UK) [22].

The quality of the sequencing was verified through the FastQC v0.11.9 program (https://github.com/s-andrews/FastQC, accessed on 18 December 2022). The Porechop v0.2.4 program [18] was used for the detection and elimination of the adapters, as well as for the demultiplexing of the Nanopore reads. Possible sequencing errors were treated with the Canu v2.1.1 monitor correction module [23]. The de novo assembly based on de Bruijn graphs of corrected sequences was carried out through the Flye v2.8.3 [24]. The contigs obtained using de novo assembly were subjected to a polishing (correction of raw contigs) with the Racon v1.4.22 program [25], which took the read mappings made with BWA v0.7.17 [26]. 

The Illumina sequencing library was prepared from genomic DNA [1 µg] using the NEBNext Fast DNA Fragmentation and Library Preparation Kit (New England Biolabs, Ipswich, MA, USA) following the manufacturer’s recommendations. The library quality was assessed using the Agilent 2100 Bioanalyzer equipment, and the paired-end DNA sequencing was carried out in the Illumina HiSeq 2500 platform. After sequencing, the raw read quality was assessed using the FastQC v0.11.5 software (https://github.com/s-andrews/FastQC, accessed on 15 January 2020). 

### 2.6. Hybrid Genome Sequence Assembly

MinION long-reads were assembled using the Racon pipeline with default parameters [24] while Illumina short reads were assembled using the (i) SPAdes version: 3.15.3 [27], (ii) Unicycler [18] and (iii) Edena [28] software with default parameters. Hybrid assemblies using Illumina and MinION reads were performed using the software (i) MaSuRCA [29], and (ii) Unicycler. Genome quality and completeness for each assembly were evaluated using QUAST v4.6.0 [30], and BUSCO v4 (Benchmarking Universal Single-Copy Orthologs) [31]. BUSCO analyses were performed using the database bacteria obd_10.

### 2.7. Serotype Identification

The identification of the serotype was carried out from the de novo contigs, using the SeqSero2 v1.2.1 program [32].

### 2.8. Gene Annotation

The annotation of genes for both the bacterial and plasmid genomes was performed through the predictor, based on hidden Markov models, Prokka v1.14.6 [33].

### 2.9. Genome Similarity Assessment

*Salmonella enterica* genomes (16,638) were downloaded from the NCBI Genbank database in July 2022. Genomes with more than 500 contigs were removed, and contigs smaller than 500 bp were removed from the remaining genomes. Genome quality was evaluated with CheckM v.1.0.13 [34], using completeness and contamination score of ≥90% and ≤10%, respectively. Genome-distance estimation of genomes was performed with Mash v.2.2.1 [35]. Near-identical redundant genomes were removed using in-house scripts to cluster genomes assemblies sharing pairwise Mash distances less than 0.005 (~99.95% average nucleotide identity (ANI)) and cluster representatives were chosen based on assembly N50. Further, the genome dataset was taxonomically verified using the Genome Taxonomy Database (GTDB). To investigate the genomic relatedness of the *S. enterica* SE3 strain and Genbank genomes, a genome-distance tree was built using a combination-distance matrix of Mash and ANI values, computed with Mash v.2.2.1 and fastANI [36], respectively. 

### 2.10. Pangenome Analysis

The *S. enterica* pangenome analysis was performed with Roary v.3.6, using 90% identity threshold to determine gene clusters [37]. The Heaps law model was used to estimate the pangenome openness. Core genes (present in up to 95% of the genomes) were aligned with MAFFT v.7.394 [38]. SNPs were extracted from the core-genome alignment using SNP-sites v.2.3.3 [39]. The phylogenetic tree was constructed using IQ-TREE [40], with ascertainment bias correction under the model GTR+ASC, and bootstrap support was performed using 1000 replicates. The resulting phylogenetic tree was visualized and rendered with iTOL v4 [41]. 

### 2.11. Mobile Genetic Element Identification and Annotation

Genomic islands were identified using Island Viewer software (www.pathogenomics.sfu.ca/islandviewer/upload/ (accessed on 18 December 2022).) [42], virulomes were detected using VFanalyser/VFDB (www.mgc.ac.cn/cgi-bin/VFs/v5/main.cgi (accessed on 18 December 2022)) [43], resistomes were identified using ResFinder-4.1 (https://cge.cbs.dtu.dk//cgi-bin/webface.fcgi?jobid=61358037000023BC9E7A4C58 (accessed on 18 December 2022)) [44], and CARD (https://card.mcmaster.ca/ (accessed on 18 December 2022)) [45], Prophages were identified using Phaster (www.phaster.ca (accessed on 18 December 2022)) [46], phage defense systems were detected using PADLOC (https://padloc.otago.ac.nz/padloc/ (accessed on 18 December 2022)) [47] and DefenseFinder (https://defense-finder.mdmparis-lab.com/ (accessed on 18 December 2022)) [48]. SPIFinder 2.0 was used to detect Pathogenic Islands (https://cge.cbs.dtu.dk/services/SPIFinder/ (accessed on 18 December 2022)) [49]. BRIG was used to draw the chromosomal *Salmonella* genomes (http://brig.sourceforge.net/ (accessed on 18 December 2022)) [50].

## 3. Results

### 3.1. Salmonella Isolation and Characterization

Presumptive *Salmonella* were isolated from soil at the Subaé River using *Salmonella* selective growth media. Isolates showed typical *Salmonella* characteristics: on XLD colonies had a slightly transparent zone of reddish color and a black center, on Bismuth Sulfite Agar there were gray or brown-black colonies with or without metallic sheen and in SS agar the colonies were beige with black centers. In biochemical tests, good growth was seen in TSI, with acid and gas reactions at depth, an alkaline surface (red) and presence of H_2_S.

### 3.2. Analysis of 16S rRNA

The presumptive *Salmonella* isolates were confirmed by 16S rRNA PCR amplification [50,51] and sequencing, followed by a sequence query of the Greengenes database. Analysis of the queries returned coverage of 100% and an E value of 0, with 99.91% identity to the same sequence, *Salmonella enterica* serovar Enteritidis (ID: MT621365.1).

### 3.3. Whole Genome Sequencing of Salmonella Isolate SE3

One of the *Salmonella* Enteritidis isolates, designated SE3, was sequenced by Illumina HiSeq and Oxford Nanopore MinION technologies. The number of reads from HiSeq sequencing was 15,997,283 and the number of reads from MinION sequencing was 13,326, after preprocessing. The MinION long reads had an average size of 5.1 kb, and the longest read was 28.8 kb (Table 1).

### 3.4. Genome Assembly

Six whole genome sequence assembly strategies, including hybrid and non-hybrid, were tested on the HiSeq and MinION sequencing data from *Salmonella* SE3 (Table 2). For Illumina HiSeq assembly, Unicycler had the best performance with 31 contigs, a total length of 4,683,367 bp, largest contig of 1,262,086 bp and N50 of 478.501 bp (Table 2). The Unicycler hybrid assembly had the best performance for genome assembly overall, with 10 contigs, total length of 4,713,463 bp, largest contig of 519,108 bp and N50 of 2,750,500 bp (Table 2) (Figure 1). When measuring genome completeness, Unicycler HiSeq and Unicycler hybrid assembly had the same result, with 98.4 % of the orthologous (complete) genes searched, 99.4 % were single-copy genes, 1.6 % genes were not identified or missing, and there were no identified single and fragmented genes (Table 3).

### 3.5. Completeness of the Genome Annotation

The genome of *Salmonella* SE3 was annotated using Prokka and rRNA, tRNA and gene coding sequences were successfully identified (Table 4 and Appendix A). *Salmonella* SE3 showed ~99.9% ANI with *Salmonella enterica* subsp. *enterica* serovar Enteritidis OLF-SE2-98984-6.

### 3.6. Genomic Relatedness of Salmonella SE3

Available *S. enterica* genomes in the GenBank database (*n* = 16,638, July 2022) were downloaded but after filtering for CheckM quality, removing highly fragmented and near-identical redundant genomes (see methods for details), the remaining dataset was 1598 genomes. Further genomic identity analysis with a combined matrix of all Mash and fastANI pairwise distances between the genomes identified a further 159 genomes with incorrect taxonomic assignment which were excluded. The distance tree built with the combined matrix showed that the *Salmonella* SE3 genome was located within the properly classified cluster of *S. enterica* genomes (Figure 2A). The *S. enterica* dataset comprised 1439 *S. enterica* genomes sharing Mash distance values up to 0.03 (~97% fastANI identity) (Figure 2B).

### 3.7. Pangenome Analysis

The pangenome of 1439 *S. enterica* genomes is composed of 74,995 gene clusters, including a core genome (present in at least 95% of the genomes) of 2137 genes. The accessory genome comprises 3390 shell or shared genes (present from 15% to 95% of the genomes) and 69,352 cloud or singletons genes (present in up to 15% of the genomes) (Figure 3B). The Heaps Law estimate supports an open pangenome (alpha = 0.52) for *S. enterica*. (Figure 3A), indicating a high genetic diversity, and the capacity of this sympatric species to rapidly acquire exogenous DNA. We also performed a maximum-likelihood phylogenetic reconstruction using 292,004 SNPs extracted from core genes. This analysis revealed that *Salmonella* SE3 belongs to a monophyletic clade containing 23 *S. enterica* strains of serovar Enteritidis (Figure 3C).

### 3.8. Genome Features

#### 3.8.1. Resistome Identification

Several resistance mechanisms were identified in *Salmonella* SE3 using the CARD database; resistance to aminoglycosides (alleles of *AAC(6’)-Iy, kdpE, baeR*), fluoroquinolones (alleles *of MdtK, emrB, emrR, sdiA, Escherichia coli acrA, acrB, rsmA, adeF*), macrolides (alleles of *Klebsiella pneumoniae* KpnE, *K. pneumoniae* KpnF, H-NS, CRP), monobactam (*golS*), nitroimidazole (*msbA*), tetracycline (*E. coli mdfA*), cephalosporin (*Haemophilus influenzae* PBP3 conferring resistance to beta-lactam antibiotics, *E. coli* EF-Tu mutants conferring resistance to Pulvomycin, *E. coli uhpT* with mutation conferring resistance to Fosfomycin, *E. coli glpT* with mutation conferring resistance to Fode novosfomycin), Figure 4.

According to their mechanism of resistance, the genes were classified as antibiotic efflux (*golS, baeR, MdtK, K. pneumoniae* KpnE, *K. pneumoniae KpnF, H-NS, sdiA, mbsA, E. coli mdfA, kdpE, E. coli acrA, acrB, adeF, CRP, rsmA, emrB, emrR* and *marA*), antibiotic inactivation (*AAC(6’)-ly*), antibiotic target alteration (*vanG, bacA, H. influenzae* PBP3 conferring resistance to beta-lactam antibiotics, *E. coli uhpT *with mutation conferring resistance to Fosfomycin, *E. coli* *EF-Tu* mutants conferring resistance to Pulvomycin, *E. coli glpT* with mutation conferring resistance to Fosfomycin, *E. coli* EF-Tu mutants conferring resistance to Pulvomycin, *pmrF, E. coli* acrAB-tolC with marR mutations conferring resistance to ciprofloxacin and tetracycline, *E. coli soxR* with mutation conferring antibiotic resistance and *E. coli soxS* with mutation conferring antibiotic resistance). Resfinder identified resistance against aminoglycosides: tobramycin (*aac(6’)-Iaa* (*aac(6’)-Iaa*_NC_003197) and amikacin (*aac(6’)-Iaa* (*aac(6’)-Iaa*_NC_003197).

#### 3.8.2. Viriome, Genomic Island and Pathogenic Island Identification

In total, 144 potential virulence genes were identified in *Salmonella* SE3 using VFanalyser/VFDB, some of the most important identified were *invA*, *sipA*, *sipB, sipC, fepA, sopA, sopB, sopD, sopE2, pefA, pefB, pefC, pefD* and *ssaB*. Genomic islands were detected using Island Viewer which uses three prediction methods: Integrated, IslandPath-DIMOB and SIGI-HMM. Twelve pathogenic islands were detected (Figure 4 and Table 5), and included virulence genes, secretion proteins, resistance genes, bacteriophage sequence regions, transposases and integrases. The gene *arsC*, encoding Arsenate reductase was identified in a genomic island. The *mdtK* gene (encoding multidrug resistance protein MdtK) was also identified in the resistome analysis. Virulence genes identified using Island Viewer were very similar to those identified using VFanalyser/VFDB.

### 3.9. Identification of Antiviral Defense Systems

Several antiviral defense system virulence genes were identified using PADLOC and DefenseFinder tools (Table 6). Both tools identified several systems: Cas type IE, CBASS type I, CRISPR array, restriction–modification (RM) RM type I, and RM type III. Similar antiviral systems and proteins were identified by PADLOC, except for AbiU and RM type II (Table 6 and Figure 4).

### 3.10. Prophage Identification

Of the prophages identified in *Salmonella* SE3 using PHASTER, two regions were intact, five regions were incomplete, and none were questionable (Table 7). Proteins were identified in the Gisfy and RE-2010 prophages including lysis, terminase, portal protein, protease, coat protein, tail shaft, attachment site, integrase, tail fiber and plate proteins.

## 4. Discussion

*Salmonella* SE3 was isolated from soil at the Subaé River in Santo Amaro, Brazil, a region contaminated with heavy metals and organic waste. The genome sequence of this isolate was determined using two sequencing technologies and six different bioinformatics strategies. Hybrid assembly showed the lowest number of contigs followed by MinION-alone assembly, with hybrid genome assembly resulting in a genome of 4.73 Mb, which was similar in size to that reported (4.68 Mb) for *Salmonella enterica* subsp. *enterica* serovar Enteritidis str. P125109 (NC_011294.1) [52]. However, the GC content of the assembled genome (52.16%) was more similar to *Salmonella enterica* subsp. enterica serovar Enteritidis str. P125109 (NC_011294.1) (52.17%) [52]. HiSeq assemblies have been traditionally considered the “gold standard” because MinION sequencing could introduce high numbers of errors and consequently may interfere with high-quality genome annotations due to reduced accuracy in gene prediction, producing a large number of misannotated genes [53,54]. However, the genome completeness of *Salmonella* SE3 with non-hybrid assembly and hybrid assembly were almost identical.

Phylogenetic analysis of the *Salmonella* SE3 genome revealed it was located within the properly classified cluster of *S. enterica*. During taxon analysis we identified 159 genomes with incorrect taxonomic classification, highlighting that it is important to confirm identity prior to undertaking phylogenetic analyses. 

The pangenome analysis of *Salmonella* SE3, revealed the core genome was composed of 2137 genes and the accessory genome comprised 3390 shell genes and 69,352 cloud genes. This indicates *Salmonella* SE3 has an open pangenome with a diversity of unique genes. A study by Chand et al. [55] undertook a comparative genomic analysis of 44 genome sequences, representing 17 serovars of *S. enterica*, and concluded that the genus *Salmonella* displays an open pangenome, comprising a reservoir of 10,775 gene families. Of these 2847 constituted the core gene families, 4657 were dispensable or accessory gene families, and 3271 strain-specific gene families. Park et al. [56] constructed pangenomes of seven species to elucidate variations in the genetic contents of >27,000 genomes, as in our study, this work showed the pangenome of *Salmonella enterica* subsp. *enterica* was open. However, it is important to note that pangenome size is heavily influenced by the properties of the genomes used and variation would likely result in inconsistencies, and secondly, newly described genes are often included which results in open pangenomes [57].

The antimicrobial resistance gene profile of *Salmonella* SE3 identified genes potentially involved in resistance to aminoglycosides, fluoroquinolones, macrolides, a monobactam (*golS*), nitroimidazole (*msbA*), tetracycline and related drugs (*mdfA*), and cephalosporins. Other studies of *Salmonella* isolates from southern Brazil have also reported tetracycline (*mdfA*) and aminoglycoside (*aac(6’)-Iaa*) resistance genes, in addition to other genes such as *aac(3)-Iva, aph(3”)-Ib, aph(4)-Ia, aph(6)-Id, tet(34)* and *tet(A)* [57,58,59,60,61]. In the United States, additional antibiotic resistance mechanisms in *S. enterica* have been described [62], such as resistance to aminoglycosides (*aadA, aadB, aacC, aphA, strAB*), β-lactams (*blaCMY-2, PSE-1*, *TEM-1*), chloramphenicol (*cat1, cat2, cmlA, floR*), inhibitors of the folate pathway (*dfr, sul*), and tetracycline (*tetA, tetB, tetC, tetD, tetG,* and *tetR*), none of these resistance genes were detected in our study. 

Ten *Salmonella* pathogenic islands were identified in *Salmonella* SE3 which is relatively high compared that reported for other *Salmonella* isolates. A *S. enterica* serovar Typhimurium isolate, ms202, from a patient in India possessed six *Salmonella* pathogenicity islands: SPI-1, SPI-2, SPI-3, SPI-4, SPI-5, and SPI-11 [63], but in our work, we did not identify SPI-4. The genes identified in SPI regions had similarity to known transporters, drug targets, and antibiotic-resistance genes, and in a subset of genomic islands, genes that facilitate the horizontal transfer of genes encoding numerous resistance and virulence factors of regions belonging to type III secretion systems (T3SS). Vilela et al. [64] analyzed six *Salmonella* Choleraesuis strains provided by the Brazilian *Salmonella* reference laboratory of the Oswaldo Cruz Foundation (FIOCRUZ-RJ), which receives *Salmonella* isolates from diverse isolation sources and regions of the country. Pathogenicity islands SPI-1, -2, -3, -4, -5, -9, -13, -14 and CS54 island were detected in five strains and SPI-11 in four strains. The majority of these SPI, with the exception of SPI 4 and SPI 11, were also detected in *Salmonella* SE3. SPI-1 and SPI-2 are known to be involved in the invasion of intestinal epithelial cells and survival and replication within phagocytic cells, respectively, through the formation of type 3 secretion systems, SPI-5 is associated with fluid secretion and inflammatory response and SPI-3, -4, -11, -13, -14 and CS54 are associated with *Salmonella* survival and adaptation to stresses within macrophages [65].

In total, 144 potential virulence genes were identified in *Salmonella* SE3. Some of these virulence genes are also found in other serovars of *Salmonella.* Borah et al. [66] investigated virulence genes in 88 *Salmonella* isolates recovered from humans and different species of animals. Among the 88 isolates, some virulence genes such *invA*, *sipA*, *sipB* and *sipC* were detected irrespective of the serovar, and these were also detected in *Salmonella* SE3. *fepA* was also present in a high percentage (64.7%) of isolates belonging to *Salmonella* serovars Enteritidis, Weltervreden, Typhi, Newport, Litchfield, Idikan and Typhimurium, as well as *Salmonella* SE3 and. Other virulence genes were present in varying percentages among the *Salmonella* serovars studied by Borah et al. [66] such as *sopB* (86.36%), *sopE2* (62.5%), *pefA* (79.54%) and *sefC* (51.14%); of these genes only *sefC* was not detected in *Salmonella* SE3. The virulence genes identified in *Salmonella* SE3 are involved in several different processes, such as the *invA* gene usually codes for a protein in the inner bacterial membrane that is responsible for the invasion of intestinal cells of the host [67,68]. The *fepA* gene encodes outer membrane receptor protein FepA, which participates in iron transport and plays a role in infection colonization in *Salmonella* [32]. T3SS-1 secretes proteins, termed effectors, across the inner and outer membranes of the bacterial cell. Some of the secreted effectors, including SipA, SipB and SipC are encoded by genes located on SPI1. The remaining effectors, including SopA, SopB, SopD, SopE and SopE2 are encoded by genes that are scattered around the *Salmonella* SE3 chromosome. Upon secretion the SipB, SipC, and SipD proteins are thought to form a complex in the eukaryotic membrane that is required for translocation of the remaining effectors into the host cell cytoplasm [69]. PefA is encoded by *Salmonella* SE3 and is the plasmid-encoded fimbrial major subunit antigen of *Salmonella* Typhimurium [70]. *Salmonella* plasmid-encoded fimbrae have been found to mediate adhesion to mouse intestinal epithelium [71].

The gene *arsC,* encoding arsenate reductase, was found in the genome of *Salmonella* SE3. Arsenate reductase is essential for arsenate resistance and transforms arsenate into arsenite which is extruded from the cell [72,73]. This is of interest as *Salmonella* SE3 was isolated from the soil of Subaé River where heavy metal concentrations were above reference values [74]. In addition, mussels (*Mytella charruana*) gathered from the same region also contained lead, arsenic and cadmium in concentrations above reference values [75]. Carvalho et al. [75] also determined the quality of soils in 39 households from nearby Santo Amaro City, and the Residential Investigation Value (RIV) was exceeded by Lead (23.1% of the samples), Cadmium (7.7%), Nickel (2.6%), Zinc (25.6%), Arsenic (2.6%), and Antimony (7.7%).

Several virus defence systems were detected in *Salmonella* SE3, including CRISPR-Cas type IE, CBASS type I, and RM type I and III systems. Similar antiviral systems and subtypes were identified by the PADLOC and DefenseFinder tools, except for AbiU and RM type II which were only identified by PADLOC. Most bacteria, including *Salmonella,* possess multiple antiviral defence systems that protect against infection by phages and mobile genetic elements [47].

Seven prophages were detected in the *Salmonella* SE3 genome, two were intact, and five were incomplete. By comparison, in *S. enterica* Typhimurium ms202 nine prophages were detected, two were intact, five were incomplete and two were questionable [63]. Moreover, *Salmonella* SE3 had not only *Salmonella* prophage sequences (e.g. phage RE-2010) but also prophages annotated as belonging to closely related genera *Shigella* (phage POCJ13) and *Escherichia* (phage 500465-2), which may indicate horizontal gene transfer or polyvalent phages. A previous study has reported that phage populations in *S. enterica* contribute to horizontal gene transfer, including virulence and virulence-related genes within the subspecies [76,77,78,79]. Further studies on *Salmonella* phages may uncover the receptor-interaction mechanisms between phages and hosts which may lead to improving phage therapy as an option for the treatment or control of *Salmonella*.

## 5. Conclusions

Salmonellosis is a healthcare issue around the world, so genomic analysis of *Salmonella* isolates could be a key determinant for better control of salmonellosis. Our study showed the effectiveness of a hybrid sequence assembly approach for environmental *Salmonella* genome analysis using HiSeq and MinION data. *Salmonella* SE3 was determined to belong to a monophyletic clade containing 23 *S. enterica* strains of serovar Enteritidis. The hybrid genome assembly enabled mobile genetic elements, genomic islands, *Salmonella* Pathogenicity Islands, antiviral systems, antimicrobial resistance genes, virulence genes, and prophages to be identified in *Salmonella* SE3. Furthermore, a gene encoding heavy metal resistance, arsC, was detected. These data are important to inform the control of *Salmonella* and heavy metal pollution in the Santo Amaro region of Brazil. 

## Figures and Tables

**Figure 1 microorganisms-11-00111-f001:**
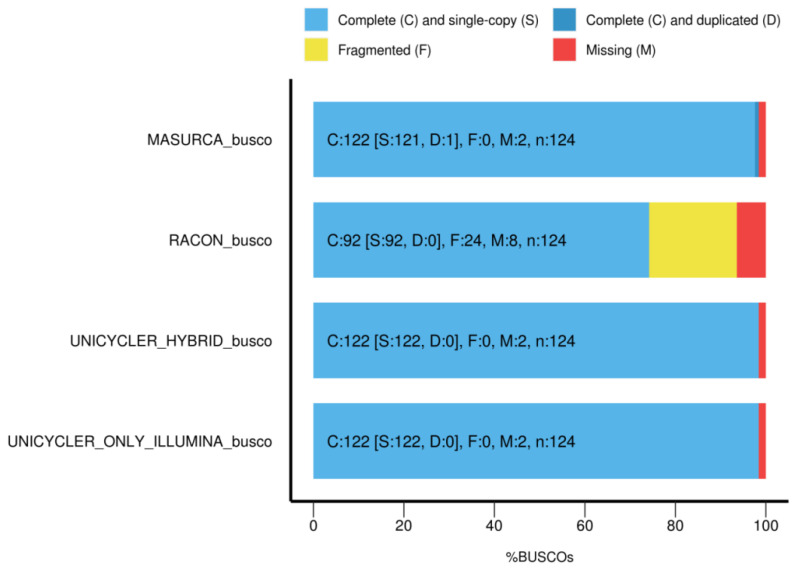
BUSCO completeness assessment of *Salmonella* SE3 genome.

**Figure 2 microorganisms-11-00111-f002:**
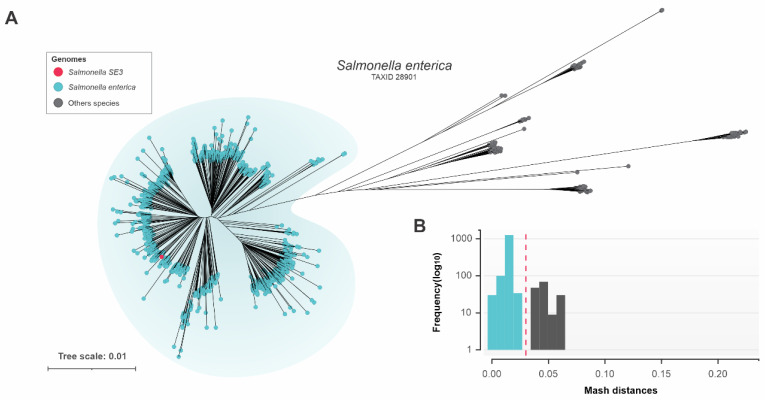
Genome similarity of *Salmonella* SE3. (**A**) Distance tree of *Salmonella enterica* built using a combined matrix of all Mash and fastANI pairwise distances of *Salmonella* SE3 and 1598 genomes. Genomes classified by GTBD as *S. enterica* are shaded in blue. (**B**) Mash-distance values of *Salmonella* SE3 were calculated with 1598 *Salmonella* genomes. The maximum Mash-distance threshold (0.03) used to select genomes is represented by a dotted line.

**Figure 3 microorganisms-11-00111-f003:**
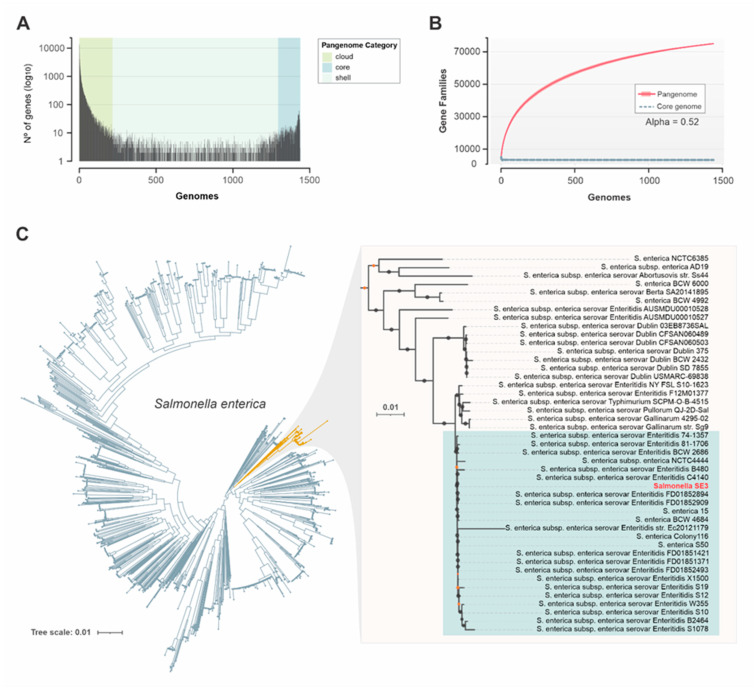
Pangenome of *Salmonella enterica* and phylogeny of *Salmonella* SE3. (**A**) Gene frequency of *S. enterica* pangenome. (**B**) Number of gene families in the *S. enterica* pangenome. The cumulative curve (in red) and an alpha value of the Heaps Law less than one (0.52) supports an open pangenome. (**C**) core-genome SNP tree of *Salmonella enterica* highlighting the phylogenetic group contained the *Salmonella* SE3 genome. The monophyletic clade containing the serovar Enteritidis of *S. enterica* is shaded in cool grey. Bootstrap values below and above 70% are represented by blue and dark-grey dots, respectively.

**Figure 4 microorganisms-11-00111-f004:**
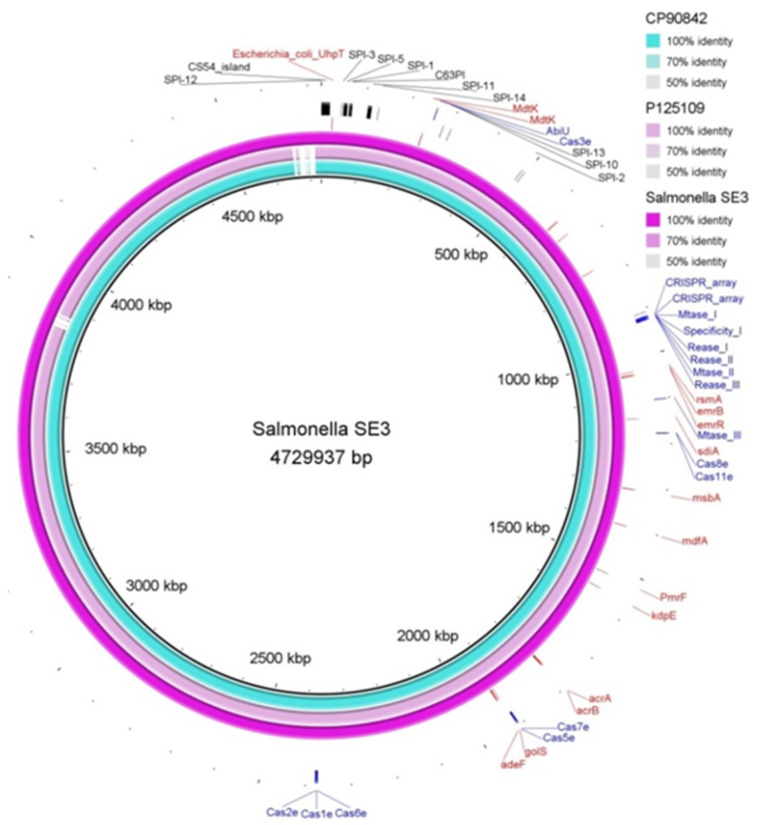
*Salmonella* SE3 antimicrobial resistance genes (red color), *Salmonella* Pathogenic Island (SP) (black color) and defense system (blue color) with two genomes of reference of *Salmonella* serovar Enteritidis (P125109 and CP9084.2).

**Table 1 microorganisms-11-00111-t001:** Summary of the Illumina HiSeq and Oxford Nanopore MinION reads statistics after preprocessing step.

Sequence Data	HiSeq	MinION
Reads	15,997,283	13,326
Total read bases (bp)	7,999,481	67,978,671
Mean coverage (%)	51,185	13,590
Longest read (bp)	151	28,841
Mean read length (bp)	150	5101
GC %	52.00	52.18
Genome size (bp)	4,688,543	4,709,033

**Table 2 microorganisms-11-00111-t002:** Summary statistics for the assembled genome of *Salmonella* SE3 using reads from Illumina HiSeq and Oxford Nanopore Technologies MinION.

Assembly Method	Racon	Unicycler	Edena	SPAdes	Unicycler	MaSuRCA
Sequence data	MinION	HiSeq	HiSeq	HiSeq	Hybrid	Hybrid
Number of contigs	2	31	41	50	10	39
Number of contigs (≥0 bp)	2	65	54	111	18	42
Number of contigs (≥50 kb)	2	15	4,475,114	4,566,140	4	24
Largest contigs	4,671,311	1,262,086	488,615	1,276,166	2,750,500	519,108
Total length (≥50 kb)	4,730,597	4,683,367	4,701,851	4,805,245	4,713,463	4,585,719
GC (%)	52.18	52.14	52.15	51.85	52.16	52.15
N50	4,671,311	478,501	181,604	491,607	2,750,500	246,991
L50	1	3	8	3	1	7

**Table 3 microorganisms-11-00111-t003:** Completeness assessment of *Salmonella* SE3 assemblies using BUSCO software.

Assembly Method	Sequence Data	Complete (%)	Single Copy (%)	Duplicated (%)	Fragmented (%)	Missing (%)
Racon	MinION	74.2	74.2	0	19.4	6.4
Unicycler	HiSeq	98.4	98.4	0	0	1.64
Unicycler	Hybrid	98.4	98.4	0	0	1.64
MaSuRCA	Hybrid	98.4	97.6	0.8	0	1.64

**Table 4 microorganisms-11-00111-t004:** *Salmonella* SE3 genome features annotated by Prokka.

Annotated Genome	Features
rRNA	20
tRNA	87
Repeat region	2
CDS	4403
mRNA	1

**Table 5 microorganisms-11-00111-t005:** Pathogenicity islands identified in *Salmonella* SE3.

No	SPI	Identity	Query/Template Length	*Salmonella* Serotype	Insertion Site	Accession Number
1	SPI-1	99.7	2705/2705	Typhimurium SL1344	*fhlA*/*mutS*	AF148689
2	SPI-2	100	642/642	Gallinarum SGC_2	tRNA-*valV*	AY956827
3	SPI-3	99.05	738/738	Typhimurium 14028s	tRNA-*selC*	AJ000509
4	SPI-5	99.11	9069/9069	Typhimurium LT2	tRNA-*serT*	NC_003197
5	SPI-10	98.28	553/554	Gallinarum SGE_3	Unpublished	AY956839
6	SPI-11	98.54	9085/15686	Choleraesuis SC_B67	Gifsy-1	NC_006905
7	SPI-12	97.14	5766/11075	Choleraesuis SC_B67	tRNA-*pro*	NC_006905
8	SPI-13	100	341/341	Gallinarum SGA_10	tRNA-*pheV*	AY956834
9	SPI-14	99.8	501/501	Gallinarum SGA_8	Unpublished	AY956835
10	C63PI	99.12	4000/4000	Typhimurium SL1344	*fhlA*	AF128999
11	CS54	98.09	19669/25252	Typhimurium ATCC_14028	*xseA*-*yfgK*	AF140550
12	Unnamed	100	330/330	Enteritidis CMCC50041	--	JQ071613

**Table 6 microorganisms-11-00111-t006:** Antiviral defense systems of *Salmonella* SE3.

Number	System	Subtype	Tool	Reference
1	AbiU	AbiU	PADLOC	[46]
2	Cas type IE	Cas3e	PADLOC	[46]
3	Cas type IE	Cas8e	PADLOC	[46]
4	Cas type IE	Cas11e	PADLOC	[46]
5	Cas type IE	Cas7e	PADLOC	[46]
6	Cas type IE	Cas5e	PADLOC	[46]
7	Cas type IE	Cas6e	PADLOC	[46]
8	Cas type IE	Cas1e	PADLOC	[46]
9	Cas type IE	Cas2e	PADLOC	[46]
10	CBASS_type_I	Cyclase	PADLOC	[46]
11	CBASS_type_I	Effector	PADLOC	[46]
12	CRISPR array	CRISPR array	PADLOC	[46]
13	CRISPR array	CRISPR array	PADLOC	[46]
14	RM type I	Mtase I	PADLOC	[46]
15	RM type I	Specificity I	PADLOC	[46]
16	RM type I	Rease I	PADLOC	[46]
17	RM type II	Rease II	PADLOC	[46]
18	RM type II	Mtase II	PADLOC	[46]
19	RM type III	Rease III	PADLOC	[46]
20	RM type III	Mtase III	PADLOC	[46]
21	Cas Class1 subtype I E1	Cas3 I 5	DefenseFinder	[47]
22	Cas Class1 subtype I E1	Cas8e I E 1	DefenseFinder	[47]
23	Cas Class1 subtype I E1	Cas2gr11 I E 2	DefenseFinder	[47]
24	Cas Class1 subtype I E1	Cas7 I E 2	DefenseFinder	[47]
25	Cas Class1 subtype I E1	Cas5 I E 3	DefenseFinder	[47]
26	Cas Class1 subtype I E1	Cas6e I II II IV V VI 1	DefenseFinder	[47]
27	Cas Class1 subtype I E1	Cas 1 I E 1	DefenseFinder	[47]
28	Cas Class1 subtype I E1	Cas2 I E 2	DefenseFinder	[47]
29	CBASS I 2	Cyclase SMODS	DefenseFinder	[47]
30	CBASS I 2	2TM Gros	DefenseFinder	[47]
31	RM Type III 2	Type III Reases	DefenseFinder	[47]
32	RM Type III 2	Type III Mtases	DefenseFinder	[47]
33	RM Type I 1	Type I S	DefenseFinder	[47]
34	RM Type I 1	Type I Mtases	DefenseFinder	[47]
35	RM Type I 1	Type I S	DefenseFinder	[47]
36	RM Type I 1	Type I Reases	DefenseFinder	[47]

MTase = Methyltransferase I, Rease = restriction endonucleases.

**Table 7 microorganisms-11-00111-t007:** Prophage sequences annotated in *Salmonella* SE3 genome.

Completeness	Score	Proteins	Position	Best Match	Accession No.	GC (%)
Incomplete	60	27	805989–831780	*Shigella* phage POCJ13	NC_025434	48.7
Intact	150	40	1041034–1072153	*Salmonella* phage Gifsy-2	NC_010393	47.2
Incomplete	50	13	1276587–1286489	*Salmonella* phage Gifsy-2	NC_010393	46.7
Incomplete	30	9	1698977–1705339	*Shigella* phage POCJ13	NC_025434	45.6
Intact	150	49	1081056–1124788	*Salmonella* phage RE-2010	NC_019488	51.2
Incomplete	20	8	1435195–1442595	*Escherichia* phage 500465-2	NC_049343	53.2
Incomplete	40	9	29216–37324	*Salmonella* phage RE-2010	NC_019488	52.4

## Data Availability

Not applicable.

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
