# Peer review of "Hybrid Genomic Analysis of Salmonella enterica Serovar Enteritidis SE3 Isolated from Polluted Soil in Brazil"

_microorganisms, 2022, doi:10.3390/microorganisms11010111_

Round 1
Reviewer 1 Report
In this study the authors describe the pangenomme of the environmental Salmonella enterica senovar enteritidis SE3 using MinION and HiSeq se quencing analysis.
1. I can understand that this combination play role to a more correct assembly but I an not sure if it has advantage regarding the determination of resistance genes, pathogenic determinants etc. Please add a comment o.n it
2. Please describe the criteria for the selection of this strain to be analyzed among the other isolates
3. Susceptibility to antimicrobial agents must be added and must be correlated with the genotype
4.The study is very descriptive but its clinical scope is not clear. The discussion is too long with repetitions of the results' findings.
5. Please correct the gene' names (italics)
Author Response
Reviewer 1
Reviewer comment
I can understand that this combination play role to a more correct assembly but I am not sure if it has advantage regarding the determination of resistance genes, pathogenic determinants etc. Please add a comment on it
Authors’ response
The effect of the sequencing / assembly method on the ability to correctly identify and annotate virulence and resistance genes was one of the things we wished to test. As per the first paragraph of the discussion “HiSeq assemblies have been traditionally considered the “gold standard” because MinION sequencing could introduce high numbers of errors and consequently may interfere with high-quality genome annotations due to reduced accuracy in gene prediction, producing a large number of misannotated genes [52,53]. However, the genome completeness of Salmonella SE3 with non-hybrid assembly and hybrid assembly were almost identical.” Once the data was analysed, as the reviewer suggests, there was no advantage in the end to any one approach, but we believe this is an important factor to consider with new sequencing technologies and bioinformatic pipelines being released regularly.
Reviewer comment
Please describe the criteria for the selection of this strain to be analyzed among the other isolates
Authors’ response
The isolates were initially identified by 16S sequencing (lines 229-233). Since they all produced a similar hit to Salmonella enterica serovar Enteritidis, one isolate was chosen for sequencing. Full whole genome sequencing of all isolates was not possible due to resource limitations. Some text has been added to make this clear (lines 234-236).
Reviewer comment
Susceptibility to antimicrobial agents must be added and must be correlated with the genotype
Authors’ response
Whilst we agree that phenotypic studies to confirm the genotype would be a useful confirmatory tool this work was undertaken in a low resource non-clinical laboratory during pandemic and the student has now graduated, so unfortunately further laboratory work of this type is not possible at this time. We have added further text (lines 468-470) to the discussion to highlight that investigation of the antibiotic susceptibility phenotype should be pursued for any future work with this organism.
Reviewer comment
The study is very descriptive but its clinical scope is not clear. The discussion is too long with repetitions of the results' findings.
Authors’ response
We have added further text (lines 536-538) to the conclusions to better highlight the clinical implications of this study. We have revised the discussion to avoid unnecessary repetition of results (lines 428-430; 443) but some data are needed for comparison with other studies.
Reviewer comment
Please correct the gene' names (italics)
Authors’ response
We have carefully reviewed the use of italics for gene names throughout the manuscript and have corrected these where necessary.
Reviewer 2 Report
Comments to the manuscript Microorganisms-2083510 entitled “Hybrid genomic analysis of Salmonella enterica serovar Enteritidis SE3 isolated from polluted soil in Brazil”.
From the presented results, it can be concluded that the authors performed a large number of experiments and essentially obtained significant results that contribute to the global understanding of the plasticity of this pathogen and to finding mechanisms of its control.
Unfortunately, several very important points are not sufficiently emphasized in order to get an impression of the importance of everything that the authors did: what advantages did the authors themselves recognize in their work of hybrid assembly, how many different isolates did they manage to identify during the work, why did they decide to sequence the SE3 strain and a lot of other information are missing that are very important.
Specific comments:
Lines 25-26. S. enterica serovar Enteritidis SE3 was isolated from river bank (riverside) soil or from the sludge (mud) of Suba in Santo Amaro, Brazil, a region contaminated with heavy metals and organic waste? The same comment should be applied later on the repeated the same statement in the text.
Line 31. … gene were encoded in the genome.
Proteins are encoded by genes, not genes are encoded. Please modify.
Silva C, Calva E, Maloy S. One Health and Food-Borne Disease: Salmonella Transmission between Humans, Animals, and Plants. Microbiol Spectr. 2014;2(1):OH-0020-2013. doi: 10.1128/microbiolspec.OH-0020-2013.
Lines 48-49. Regarding information that Salmonella enterica subspecies enterica has more than 2600 serovars citations are better:
WHO Collaborating Centre for Reference and Research on Salmonella. Antigenic formulae of the Salmonella serovars. 9edn, (1997).
S, Tille A, Fruth, Flieger A. Genome-based Salmonella serotyping as the new gold standard. Sci Rep 10(1), 4333 (2020). https://doi.org/10.1038/s41598-020-61254-1
Line 84. ….bridges to resolve all repeated sequence in the genome, performs multiple rounds of short-read pol…….
Lines 92-93. Please be more precise from where sample(s) was taken, From river, riverside, mud, land close to the river, how many meters distance from river soil sample was taken.
Line 95. …. box (4–8°C) to the laboratory where the analyses were undertaken immediately.
What analysis was performed (chemical analysis for heavy metals) and other chemical composition, bacterial composition and so on, or isolation of bacteria from the sample was immediately undertaken.
Lines 98-99. Briefly, 10 g or 10 mL of samples of each sample were pre-enriched in 100 mL lactose broth (supplier),…….
Does this mean that a larger number of samples were collected given that we have liquid and solid material or was the general procedure cited. However, only what was done should be stated.
Lines 101- 104. Broth cultures from the selective enrichment broth were plated on Xylose-Lysine-Desoxycholate…..
Please indicate how much “One hundred microliters of enriched culture were plated on solid medium……”
Lines 108-110. Isolates that showed good to excellent growth, pink-coloured colonies with black centers on red agar were considered as Salmonella.
Lines 112-114. For bacteria, a single colony was enriched in 5 ml Luria Bertani (LB) broth, and 15 mL of enrichment broth was transferred to a centrifuge tube and centrifuged at 4000 rpm for 10 min.
How is it possible to inoculate only 5 ml and use 15 ml culture for DNA isolation (where did the other 10 ml come from)?
Line 116. For phages, a crude lysate was centrifuged the lysate as described.
Where it was described? In addition, “lysate” was repeated two times.
Lines 124-127. PCR amplification was performed using a VeritiTM 96-Well Thermal Cycler (Applied Biosystems, Foster City, CA), 16S gene Amplification PCR for the amplification of the 16S rRNA_ gene was carried out using universal primers 27F (5′-AGAGTTTGATCATGGCTCAG-3′) as forward and 1492R (5′-GGTTACCTTGTTACGACTT-3′) as a reverse primer [20].
These are two sentences, please split it into two. In addition, I don't understand what reference 20 is referring to, whether it's the primers (which is not correct) or the previous how someone did it.
Lines 133-134. PCR products were visualized using GelRed 133 (Biotium, San Francisco, CA) on a 2% agarose gel which had been run at 80 V for 30 min.
Why on 2% agarose gel since PCR fragment is 1.5 kB, better 1% agarose
Line 137. The amplified 16S rRNA gene PCR products were purified……
Lines 186-187. Genomes with more than 500 contigs were removed, and contigs smaller than 500 bp were removed from the remaining genomes.
I don't understand to what this refers to, whether it's on the downloaded genomes or the sequenced genomes in this study; should be clearer.
Line 228. 3.2 16S Analysis
16S is not enough, Please, write full names 16S rRNA gene analysis. Also, later.
Line 229. I don't understand what the quote has to do with confirming the isolate's identity as Salmonella via 16S rRNA gene sequencing.
Lines 234-235. One of the Salmonella Enteritidis isolates, designated SE3, was sequenced by Illumina HiSeq and Oxford Nanopore MinION technologies
A logical question arises here: How many different isolates were selected, on the basis of which a given strain SE3 was selected for sequencing, etc. These are all important information, why that one.
Lines 408-409. However, the genome completeness of Salmonella SE3 with nonhybrid assembly and hybrid assembly were almost identical.
So, what is the advantage of the hybrid approach, should be point out since authors used it to complete genome sequence of their isolate SE3?
What I miss in the results is: what results were obtained regarding the isolated bacteriophage DNA. Why was bacteriophage DNA isolated? Whether plaques were obtained in the essay at all, whether induction of prophages with mitomycin C was done, that is not entirely clear to me. Maybe that the authors wanted to see which prophages could be induced and compared to the sequence in the genome.
Regarding to importance of arsenate reductase beside (71. Jackson et al. 2003) reference
Ben Fekih I, Zhang C, Li YP, Zhao Y, Alwathnani HA, Saquib Q, Rensing C, Cervantes C. Distribution of Arsenic Resistance Genes in Prokaryotes. Front Microbiol. 2018;9:2473. doi: 10.3389/fmicb.2018.02473. is adequate.
Author Response
Reviewer 2 Comments
Reviewer comment
Lines 25-26. S. enterica serovar Enteritidis SE3 was isolated from river bank (riverside) soil or from the sludge (mud) of Suba in Santo Amaro, Brazil, a region contaminated with heavy metals and organic waste? The same comment should be applied later on the repeated the same statement in the text.
Authors’ response
The sample was from the riverbank soil, not from the mud. This has been updated in the document in various places.
Reviewer comment
Line 31. … gene were encoded in the genome.
Proteins are encoded by genes, not genes are encoded. Please modify.
Authors’ response
We have changed “encoded” to “present” in this text (line 31).
Reviewer comment
Lines 48-49. Regarding information that Salmonella enterica subspecies enterica has more than 2600 serovars citations are better: WHO Collaborating Centre for Reference and Research on Salmonella. Antigenic formulae of the Salmonella serovars. 9edn, (1997). S, Tille A, Fruth, Flieger A. Genome-based Salmonella serotyping as the new gold standard. Sci Rep 10(1), 4333 (2020). https://doi.org/10.1038/s41598-020-61254-1. Silva C, Calva E, Maloy S. One Health and Food-Borne Disease: Salmonella Transmission between Humans, Animals, and Plants. Microbiol Spectr. 2014;2(1):OH-0020-2013. doi: 10.1128/microbiolspec.OH-0020-2013.
Authors’ response
Thank you for the suggested references. These references have replaced the existing #5, #6 and #7 references in the manuscript.
Reviewer comment
Line 84. ….bridges to resolve all repeated sequence in the genome, performs multiple rounds of short-read pol…….
Authors’ response
We have revised this section to make it clearer (lines 84-85)
Reviewer comment
Lines 92-93. Please be more precise from where sample(s) was taken, From river, riverside, mud, land close to the river, how many meters distance from river soil sample was taken.
Authors’ response
We have clarified that the sample was taken from the riverbank “approximately 1 m from the main water body” (lines 96-97)
Reviewer comment
Line 95. …. box (4–8°C) to the laboratory where the analyses were undertaken immediately. What analysis was performed (chemical analysis for heavy metals) and other chemical composition, bacterial composition and so on, or isolation of bacteria from the sample was immediately undertaken.
Authors’ response
Isolation of Salmonella from the sample was immediately undertaken. We have made this clearer (line 98).
Reviewer comment
Lines 98-99. Briefly, 10 g or 10 mL of samples of each sample were pre-enriched in 100 mL lactose broth (supplier),…….Does this mean that a larger number of samples were collected given that we have liquid and solid material or was the general procedure cited. However, only what was done should be stated.
Authors’ response
We have amended the protocol to remove the liquid sampling which was not performed on this occasion.
Reviewer comment
Lines 101- 104. Broth cultures from the selective enrichment broth were plated on Xylose-Lysine-Desoxycholate…..Please indicate how much “One hundred microliters of enriched culture were plated on solid medium……”
Authors’ response
The volume plated (10µl) has been added to the method (lines 112-113)
Reviewer comment
Lines 108-110. Isolates that showed good to excellent growth, pink-coloured colonies with black centers on red agar were considered as Salmonella.
Authors’ response
We have made the edits as suggested (line 120)
Reviewer comment
Lines 112-114. For bacteria, a single colony was enriched in 5 ml Luria Bertani (LB) broth, and 15 mL of enrichment broth was transferred to a centrifuge tube and centrifuged at 4000 rpm for 10 min.
How is it possible to inoculate only 5 ml and use 15 ml culture for DNA isolation (where did the other 10 ml come from)?
Authors’ response
A single colony was enriched in 5 ml broth, then 1 ml of the culture was transferred to 15 mL of fresh broth prior to centrifugation. We have updated the text to make this clear (lines 123-124).
Reviewer comment
Line 116. For phages, a crude lysate was centrifuged the lysate as described.
Where it was described? In addition, “lysate” was repeated two times.
Authors’ response
This line was included by mistake, the isolation of phages / phage DNA was not included in this work, so it has been deleted (line 122).
Reviewer comment
Lines 124-127. PCR amplification was performed using a VeritiTM 96-Well Thermal Cycler (Applied Biosystems, Foster City, CA), 16S gene Amplification PCR for the amplification of the 16S rRNA_ gene was carried out using universal primers 27F (5′-AGAGTTTGATCATGGCTCAG-3′) as forward and 1492R (5′-GGTTACCTTGTTACGACTT-3′) as a reverse primer [20]. These are two sentences, please split it into two. In addition, I don't understand what reference 20 is referring to, whether it's the primers (which is not correct) or the previous how someone did it.
Authors’ response
The sentence has been split as requested (line 134). Reference 20 refers to the method and primer sequences. The primers were originally by Lane 1991 and Turner 1999, respectively, but we feel it is more appropriate to cite the publication from which the methods were followed rather than just the primer sequences (which are cited within Ref #20). We have made this clearer (line 136).
Reviewer comment
Lines 133-134. PCR products were visualized using GelRed 133 (Biotium, San Francisco, CA) on a 2% agarose gel which had been run at 80 V for 30 min.
Why on 2% agarose gel since PCR fragment is 1.5 kB, better 1% agarose
Authors’ response
The visualisation of the pcr products and ladder was best in our system using 2% agarose.
Reviewer comment
Line 137. The amplified 16S rRNA gene PCR products were purified……
Authors’ response
We have corrected all instances of “16S” to be “16S rRNA”.
Reviewer comment
Lines 186-187. Genomes with more than 500 contigs were removed, and contigs smaller than 500 bp were removed from the remaining genomes. I don't understand to what this refers to, whether it's on the downloaded genomes or the sequenced genomes in this study; should be clearer.
Authors’ response
It is referring to the downloaded genomes. This has been reworded for clarity (lines 216-217).
Reviewer comment
Line 228. 3.2 16S Analysis. 16S is not enough, Please, write full names 16S rRNA gene analysis. Also, later.
Authors’ response
We have corrected all instances of “16S” to be “16S rRNA”.
Reviewer comment
Line 229. I don't understand what the quote has to do with confirming the isolate's identity as Salmonella via 16S rRNA gene sequencing.
Authors’ response
The reference highlights the potential issues with 16s rRNA PCR amplification and need for qualification, such as sequencing, to confirm the amplicon is not contaminated with other sequences.
Reviewer comment
Lines 234-235. One of the Salmonella Enteritidis isolates, designated SE3, was sequenced by Illumina HiSeq and Oxford Nanopore MinION technologies. A logical question arises here: How many different isolates were selected, on the basis of which a given strain SE3 was selected for sequencing, etc. These are all important information, why that one.
Authors’ response
The isolates were initially identified by 16S sequencing (lines 229-233). Since they all produced a similar hit to Salmonella enterica serovar Enteritidis, one isolate was chosen for sequencing. Full whole genome sequencing of all isolates was not possible due to resource limitations. Some text has been added to make this clear (line 234-236).
Reviewer comment
Lines 408-409. However, the genome completeness of Salmonella SE3 with nonhybrid assembly and hybrid assembly were almost identical. So, what is the advantage of the hybrid approach, should be point out since authors used it to complete genome sequence of their isolate SE3?
Authors’ response
The effect of the sequencing / assembly method on the ability to correctly identify and annotate virulence and resistance genes was one of the things we wished to test. As per the first paragraph of the discussion “HiSeq assemblies have been traditionally considered the “gold standard” because MinION sequencing could introduce high numbers of errors and consequently may interfere with high-quality genome annotations due to reduced accuracy in gene pre-diction, producing a large number of misannotated genes [52,53]. However, the genome completeness of Salmonella SE3 with non-hybrid assembly and hybrid assembly were almost identical.” Once the data was analysed, as the reviewer suggests, there was no advantage in the end to any one approach, but we believe this is an important factor to consider with new sequencing technologies and bioinformatic pipelines being released regularly.
Reviewer comment
What I miss in the results is: what results were obtained regarding the isolated bacteriophage DNA. Why was bacteriophage DNA isolated? Whether plaques were obtained in the essay at all, whether induction of prophages with mitomycin C was done, that is not entirely clear to me. Maybe that the authors wanted to see which prophages could be induced and compared to the sequence in the genome.
Authors’ response
Please refer to the earlier comment. Isolation of virulent and temperate phages were not included in the analysis.
Reviewer comment
Regarding to importance of arsenate reductase beside (71. Jackson et al. 2003) reference
Ben Fekih I, Zhang C, Li YP, Zhao Y, Alwathnani HA, Saquib Q, Rensing C, Cervantes C. Distribution of Arsenic Resistance Genes in Prokaryotes. Front Microbiol. 2018;9:2473. doi: 10.3389/fmicb.2018.02473. is adequate.
Authors’ response
Thank you for the suggestion. We have replaced reference #71 with this reference.
Round 2
Reviewer 2 Report
Authors accepted almost all suggestions and changed manuscript accordingly, that improved it considerably.